# Development of a Psychological Intervention to Improve Depressive Symptoms and Enhance Adherence to Antiretroviral Therapy among Adolescents and Young People Living with HIV in Dar es Salaam Tanzania

**DOI:** 10.3390/healthcare10122491

**Published:** 2022-12-09

**Authors:** Tasiana Njau, Fileuka Ngakongwa, Bruno Sunguya, Sylvia Kaaya, Abebaw Fekadu

**Affiliations:** 1Department of Psychiatry and Mental Health, The Muhimbili University of Health and Allied Sciences, 9 United Nations Road, Upanga West 11103, Dar es Salaam 65001, Tanzania; 2Centre for Innovative Drug Development and Therapeutic Trials for Africa (CDT-Africa), Addis Ababa University, Addis Ababa 9086, Ethiopia; 3Department of Psychiatry, School of Medicine, College of Health Sciences, Addis Ababa University, Addis Ababa 9086, Ethiopia; 4Department of Psychiatry and Mental Health, Muhimbili National Hospital, Dar es Salaam 65000, Tanzania; 5Department of Community Health, Muhimbili University of Health and Allied Sciences, 9 United Nations Road, Upanga West 11103, Dar es Salaam 65001, Tanzania; 6Department of Global Health & Infection, Brighton and Sussex Medical School, Brighton BN1 9PX, UK

**Keywords:** psychological intervention, behavioral intervention, adolescent depression, ART adherence, individual therapy, theory of change, intervention development

## Abstract

**Background**: Interventions that simultaneously target depression and antiretroviral therapy (ART) medication adherence are recommended for improving HIV treatment outcomes and quality of life for adolescents living with HIV. However, evidence is scarce on culturally feasible and acceptable interventions that can be implemented for HIV-positive adolescents in Tanzania. We, therefore, developed a manualized brief psychological intervention that utilizes evidence-based strategies to address depression and ART adherence in adolescents living with HIV in Tanzania. **Methods**: We used the Theory of Change Enhanced Medical Research Council framework (TOCMRC) for developing complex interventions in health care to develop the intervention in five phases. First, the literature was reviewed to identify potential intervention components. Second, we conducted a situational analysis using qualitative interviews with adolescents living with HIV, health care providers, and caregivers. Third, we conducted a mental health expert workshop; and fourth, theory of change workshops with representatives from the Ministry of Health, mental health professionals, HIV implementing partners, adolescents, and healthcare providers. Lastly, we synthesized results to finalize the intervention and a theory of change map showing the causal pathway for how we expect the developed intervention to achieve its impact. **Results:** Adolescents living with HIV in Tanzania experience several unmet mental health needs ranging from overwhelming depressive symptoms to not feeling understood by healthcare providers who lack mental health knowledge. Participants perceived psychological intervention that utilizes a task-shifting approach to be acceptable and beneficial to addressing those problems. The novel components of the NITUE intervention included incorporating evidence-based intervention components, namely, cognitive–behavioral therapy, motivational interviewing, and problem solving. In addition, caregiver inclusion in the treatment was essential to ensure access to care, compliance, and improved outcomes. **Conclusions:** A culturally appropriate brief psychological intervention that utilizes a task-shifting approach to address depression and medication adherence for adolescents living with HIV in Dar es Salaam, Tanzania, was developed. The intervention will be piloted for appropriateness, feasibility, and acceptability and will provide material for a future trial to determine its effectiveness.

## 1. Introduction

About 38.4 million people live with Human Immunodeficiency Virus (HIV) worldwide, and 18.8 million are adolescent girls and young women [1,2]. Although new HIV infections in sub-Saharan Africa have decreased from 1.9 million in 2010 to 1.7 million in 2019, the prevalence of HIV in adolescents remains high [3]. About 3.8 million adolescents between 15 and 24 years were living with HIV in sub-Saharan Africa, equating to 76% of the global burden of the disease among young people [4,5]. Like other countries in the region, Tanzania is also experiencing a high burden of HIV among adolescents and young adults [3,6]. With a third of its population comprising adolescents and young people between 10–24 years old [7], nearly half of new HIV infections occur in this group [5]. Hence, Tanzania carries 5% of the global burden of HIV among adolescents [1,5].

Adolescents living with HIV (ALWHIV) have an increased risk for depression, negatively affecting ART adherence [8,9,10]. In Tanzania, the prevalence of depression among ALWHIV is 47.1% [11,12], and despite such a high burden, depression remains underdiagnosed and undertreated. Management of depression ensures better adherence to ART medication and hence better treatment outcomes, as well as improved quality of life [12,13,14,15].

Evidence suggests that single-approach interventions such as medication alone are inadequate in resolving depression and its associated difficulties, such as adherence to ART medication for adolescents with co-morbid HIV and depression. Adhering to anti-depressants has also proved challenging because of their interactions with ART and the additional tablet burden; hence, including psychological interventions alongside medical treatment is essential [16]. For adolescents, the mental health Gap Action Program (mhGAP) recommends brief psychological interventions as a first-line treatment for common mental disorders such as depression [17].

Studies in Tanzania have widely recommended integrating mental health services and adherence interventions into the care and treatment of adolescents living with HIV [11,13,18,19,20]. A cause for alarm is that, even with a significant amount of work being carried out to support adolescents living with HIV/AIDS [21], there is a scarcity of data and evidence-based interventions that address both depression and ART adherence in adolescents living with HIV in Tanzania and SSA [9,20].

Studies that have identified effective interventions to improve depression and ART adherence in well-resourced countries suggest that cognitive–behavioral therapy is an effective treatment for depression [18,22,23]. Moreover, the client-centered approach in motivational interviewing [24] is suitable for addressing the complex multidimensional features of ART adherence [24,25,26]. Research on brief psychological interventions that combined the two approaches (CBT and MI) to address depression and medication adherence in ALWHIV has recently increased in high-resourced countries and with positive findings [25,27,28,29]. Similar interventions are encouraged to be scaled up in Tanzania even when they have not been developed or adapted for use in this poor-resourced setting [30,31,32].

The issues on the global mental health agenda have raised concerns about the cultural appropriateness of interventions taken to scale in low and middle-income countries LMIC [33]. The arguments highlight the challenges and risks of applying Western approaches and interventions to other cultures prior to assessing if inherent are socially appropriate, feasible, or effective [26,34,35]; hence, intervention development or good adaptation work needs consideration before going to scale. The aim is to ensure that the interventions can deliver care of acceptable standard and that the strategies are replicable [36,37] in these settings where human resources are limited [38,39]. The current focus of Global Mental Health and the World Health Organization (WHO) in addressing the gap in treating mental health problems, especially for ALWHIV in LMIC, recommends evidence-based interventions that can be integrated and scaled up through task shifting to improve access to service [34].

To fill that gap, we developed a brief psychological intervention that utilizes a task-shifting approach to address depression and ART medication adherence for ALWHIV in Dar es Salaam, Tanzania.

## 2. Methods

### 2.1. Design

We used the Theory of Change enhanced Medical Research Council framework (ToCMRC) for developing complex interventions in healthcare [40,41] to guide the development process. Iteratively, we followed the four phases of the MRC framework (intervention development, feasibility and piloting, evaluation, and implementation). The ToC map provides a graphic representation of the causal pathways through which the developed intervention is expected to achieve its impact. The map includes (i) the final outcome, which is the primary intervention outcome (improvement in depressive symptoms and enhanced ART adherence); (ii) intermediate outcomes, which include things that are currently unavailable, and that need to be there to achieve the outcome; (iii) interventions which are the different components of the intervention that are needed to move from one outcome to another; (iv) assumptions which are the conditions that are beyond the intervention that needs to be there to move forward in achieving the outcome); (v) the rationale from evidence or experience that explains the pathway for each outcome; and (vi) indicators which are measurable things that determine whether the outcome has been achieved or that there is progress toward aching the outcome [40]. The ToCMRC framework was favored because it provides a basis for enhanced stakeholder engagement that ensures specific intervention components will be well set within the local Tanzania cultural context [40]. The framework also systematically identifies knowledge gaps and clearly describes pathways to the outcomes of interest, in this case, reducing depressive symptoms and enhancing ART adherence. Additionally, with this framework, information obtained during the piloting process can be used to improve the intervention [40,42,43].

### 2.2. Intervention Development Process

Intervention development work started with a review of literature that facilitated the identification of potential components of the current intervention and a description of how those components could improve depressive symptoms or enhance ART adherence [44,45,46]. We also conducted a situational analysis comprising public data on adolescents living with HIV, treatment coverage, and integration of mental health services in Dar es Salaam. These data were obtained from Management and Development Health (MDH), a non-profit organization that supports HIV care and treatment services delivery in Tanzania, the District Medical Officer, and observation of service provision in selected centers that provide adolescent HIV care and treatment in Dar es Salaam. Furthermore, qualitative surveys were carried out to determine the unmet mental health needs of adolescents living with HIV, and the barriers and opportunities for implementing an integrated psychological intervention described elsewhere [47,48]. The qualitative study’s findings shed light on lived experiences and indicate the problems from the perspectives of adolescents, caregivers, and healthcare providers. Qualitative findings also facilitated the identification of psychosocial factors that could be considered in the current intervention.

The development process engaged mental health professionals such as psychiatrists and clinical Psychologists from the Muhimbili University of Health and Allied Sciences (MUHAS) and the Muhimbili National Hospital (MNH). We also collected views on developing the intervention from the adolescents, caregivers, and health care providers in HIV care and treatment centers. Furthermore, essential stakeholders from the government, such as the district medical officer, the ministry of health, and HIV implementing partners in Tanzania, were also involved as they directly affect the implementation of interventions. The process ensured these stakeholders had a picture of possible outcomes within the existing structure of CTCs in Dar es Salaam.

### 2.3. Setting

The study was conducted in Dar es Salaam, the business capital city of Tanzania, with five districts: Ilala, Kinondoni, Ubungo, Temeke, and Kigamboni. The qualitative study was carried out at the HIV care and treatment centers in the Kinondoni district because these centers provide HIV services for adolescents on a different day than adults and have providers who may be more likely to understand the dynamics of HIV and mental health in ALHIV. MDH facilitated the selection of HIV care and treatment centers. The expert meeting and ToC workshops were conducted in the ilala district, where the Muhimbili University of Health and Allied Sciences and the Muhimbili National Hospital is located.

## 3. Study Participants and Sampling

We included individuals aged 18 years and older, able to provide written, informed consent to study participation. Participants also had to be either a mental health professional, a key stakeholder from the government/Ministry of Health, an HIV implementing partner, or an HIV healthcare provider with experience working in the field or directly with ALWHIV. The study also included ALWHIV who were at least 18 years old and their caregivers (parents or guardians). We used a purposive sampling approach [49] to recruit a diverse sample of participants. We considered those with experience or expertise in adolescent HIV and mental health care. Adolescents were recruited from the three selected clinics with the help of a healthcare provider from the respective clinic who was familiar with adolescents who can communicate, are physically and mentally stable, and are aware of their HIV-positive status.

## 4. Mental Health Professionals’ Workshop

Two mental health professional meetings were conducted. The first included twenty-nine [29] mental health professionals (four psychiatrists, five clinical psychologists, three social workers, four psychiatric nurses, and thirteen masters level residents in psychiatry and clinical psychology). First, findings from phase one of the study, which consisted of the literature review and the qualitative studies, were presented and discussed. Next, mental health professionals discussed and gave recommendations on the content and delivery of the intervention based on the proposed findings and their clinical and research experience. The workshop was conducted in Swahili and English at the Muhimbili National Hospital’s Department of Psychiatry and Mental Health and lasted two hours. Three people took notes for the meeting. The second meeting was between two psychiatrists and three psychologists from the first meeting, who reviewed inputs submitted in writing by participants from the first meeting. Combined notes were taken to come up with one document.

## 5. Theory of Change Workshops

Four ToC meetings were conducted each for a half day with the Ministry of Health and HIV implementing partners and district mental health officer (four), HIV health care providers (twelve), adolescents living with HIV (eight), representatives from implementing partners (two), health care providers (four and adolescents living with HIV (four)) We purposefully recruited the participants to include those more likely to give their opinion and consider their experience in supporting or delivering HIV and mental healthcare. All ToC workshops were facilitated by TN, a master-level clinical psychologist with intensive training in ToC. Efforts were made to ensure all participants got an opportunity to participate in each component fully. Summaries of key points were projected on a PowerPoint presentation or laid out in flip charts and revised after each subsection to validate the recorded information. After the workshop, using notes recorded during the workshop by two research assistants and flipcharts from the seminar, the Theory of Change map was updated.

The three TOC meetings started with a presentation that described the objectives of the meeting, which were (a) to explore the feasibility and acceptability of a psychological intervention to address depression in ALWHIV, and (b) to develop a ToC map that will indicate the causal pathway through which the newly developed psychological intervention was expected to achieve its impact. The presentation also covered findings from the literature review and qualitative studies. English and or Kiswahili were used as the medium of communication. After the introductory presentation, a discussion was led by TN encouraging participants to discuss the findings with a focus on possible social-cultural benefits of psychological intervention for adolescents living with HIV. Participants were also encouraged to consider the feasibility of such intervention within the existing structure and the delivery of HIV care in Tanzania, including human resources, time, and other possible barriers.

Discussion of the various components of ToC started by asking participants what they would like as the outcome of the psychological intervention for HIV-positive adolescents with depressive symptoms and challenges in ART medication adherence. We used sticky notes to map things participants discussed as unavailable but needed to be there to achieve the desired outcomes. Next, we used an example of growing a home shade tree to help participants discuss intermediate activities and conditions that need to be there to move from one intermediate outcome to the other, indicators of outcomes, and measurable things that determines whether the outcome has been achieved or there is progress toward achieving the outcome. The example included questions like; what are the desired results of planting a tree in the garden? What needs to be conducted before planting a tree, and why the preparatory process is necessary for obtaining the desired effect/outcome? The tree example was beneficial, especially for adolescent ToC meetings, as their response made it easy to relate the outcome assumptions and interventions used to develop the psychological intervention.

For each ToC meeting, the ToC map for each session was drafted by reviewing workshop minutes and notes. An integrated ToC map was then developed from the three maps. Finally, it was presented in the fourth Toc workshop that included representatives from each group in the previous ToC meetings to come up with a final agreement on the content, length, and number of sessions, and strategies to overcome potential barriers in the implementation of integrated psychological services for adolescents living with HIV. Next, TN and two other psychologists did a two-day intervention planning workshop to refine the ToC map and finalize the content of the psychological intervention. Then, before reviewing and approving the manual for pilot testing, the draft manual was translated into Swahili.

## 6. Analysis

Following Graneheim and Lundman [50], qualitative content analysis guided data analysis. Two researchers analyzed the data to ensure reliability [51]. All Toc meeting transcripts, flip charts, and notes were first to read and re-read by two authors (TN and FN). The qualitative data analysis with NVivo software was used to manage and organize data. Condensed meaning units related to participants’ description of components of the ToC map, including outcomes, intervention, preconditions, assumptions and indicators, acceptability, and feasibility, were formed through data reduction. Since codes were pre-defined by the ToCMRC frameworks, the authors agreed on the main categories by checking the similarities and differences of sub-categories and reflecting upon the interpretations of the participant’s descriptions. Quotes were selected to support the presented themes and categories. Although the description seems to be linear, the analysis process was iterative. Through discussion, ToC maps were developed. The authors then combined the ToC maps from each workshop and devised a single ToC map. The final draft was refined and approved by the last workshop, which consisted of representatives of participants from the previous three ToC workshops.

## 7. Data Triangulation

A triangulation of findings from different sources was conducted to ensure the trustworthiness of this study. Credibility was ensured by collecting data from adolescents, health care providers, and other essential stakeholders to provide comprehensive accounts of their experiences, establish buy-in of the intervention in the health care system and delivery, as well as to ensure its content and structure are well rooted in the local situation [52,53]. At least two authors conducted the initial analysis, and all authors reviewed and agreed on categories. Data accuracy was validated through summaries for participants’ checks and the final meeting that reviewed and approved the last ToC map [54]. To increase the study’s credibility, we included participants’ quotes to support the results [55]. Two senior researchers (AF and SK) closely monitored, reviewed, and examined the research process and data analysis to ensure consistent findings to promote dependability and conformability. Finally, the Guidance for Reporting Intervention Development (GUIDED) checklist [56] was used to report the intervention development process.

## 8. Ethical Considerations

The Muhimbili University of Health and Allied Sciences IRB granted ethical approval for the study in Dar-es-Salaam, Tanzania (Ref. No. DA.282/298/01.C/053). Additionally, the study was approved by the Addis Ababa University Institutional Review Board (Ref. No. 051/20/CDT). The Kinondoni Reginal Medical officer permitted the conduct of the study. Before the study, written informed consent to participation, a record of discussion, and findings used for publication were obtained for all study participants at least 18 years. In total, 2 participants were below 18 years and written parental permission and their accent were also obtained.

## 9. Results

### 9.1. Mental Health Expert Workshop

The expert workshop suggested that the intervention be delivered in six individual sessions. Participants argued that the intervention should be designed to address underlying psychological problems/stressors specific to adolescents living with HIV, as identified in the qualitative study. Psychoeducation was proposed to be included briefly, focusing on illness/symptoms and available treatment options. Mental health professionals suggested that nurse counselors be trained in communication skills to help them burst common myths about mental health problems, change attitudes, and reduce stigma. Communication skills were thought to be necessary to improve and build therapeutic relationships and improve assessment and management skills that will enhance the acceptability of the intervention.

The mental health professionals further highlighted the importance of utilizing evidence-based approaches from cognitive behavioral therapy and motivational interviewing, as supported by the literature [10,45,57,58]. It was argued that the cognitive–behavioral component would be helpful for adolescents to understand and practice how thoughts affect emotional behaviors and change maladaptive thoughts. It was, however, discussed and agreed that homework might not be very acceptable to adolescents; hence they should be limited to more practical activities such as relaxation exercises, mood monitoring, and behavioral activation. In addition, motivational interviewing strategies were thought necessary to enhance and promote healthy behaviors and motivation to take care of their health, adapt to their HIV status, and adhere to the treatment regime.

Mental health professionals suggested that including problem solving will make the intervention more culturally acceptable. It was agreed that problems identified in the qualitative data that required generating solutions, such as dealing with stigma and improving academic performance, could be solved by improved problem-solving skills. Mental health professionals suggested that the Intervention sessions last for approximately 30–45 min. The mental health professionals further recommended that the intervention manual be used flexibly but should provide the provider using it with guidelines on the content and structure of the session. It was discussed and agreed that the provider can, however, modify a session based on their clinical judgment if the overall approach is consistent with the principles of the intervention.

The expert suggested that one or two additional sessions may be included in the manual and used throughout the interventions to address specific concerns of the adolescent receiving the intervention or the provider. The supplement areas were suggested to be designed to assist the counselor in applying core skills to common psychological problems and stressors for adolescents living with HIV, such as the loss of a significant other, stigma, and violence due to HIV status.

### 9.2. ToC Workshops

Three ToC workshops and one ToC workshop that included the representative of all participants were conducted, with a total of twenty-two participants, as indicated in Table 1.

The fourth ToC meeting included representatives from the three workshops purposefully selected to meet representation requirements. The synthesized findings were reviewed, and TN and FN drafted the intervention manual that all authors critically reviewed.

### 9.3. Intervention Development Recommendations

Participants perceived that a psychological intervention was needed to address depression and associated problems in adolescents living with HIV. They, however, indicated the importance of ensuring feasibility, acceptability, and applicability within the local context.

Caregivers of adolescents insisted on using local terminologies and other easy-to-understand and acceptable approaches to helping adolescents and caregivers understand depression. Local idioms were vital as they will facilitate understanding, health help-seeking behavior, coping strategies, and the intervention’s probability of bringing about the intended effect.

“*You will need to use terminologies that we understand, do not say depression because most will not understand. Better use a simple description like overthinking and give some more description of the symptom*.”

The intervention was perceived as necessary because it would provide a solution to many adolescents’ problems and reduce the time spent by caregivers looking for help.

“*It is hard to imagine that somewhere in this area has the solution to the problem that we parents have been unsuccessfully finding. Availability of treatment in the CTC will save us the movement from church, traditional healers, and so many places that have not been helpful in this*.”

The HCP thought that the intervention would be acceptable for nurse counselors targeted to deliver the intervention. In addition, they suggested that management manuals and screening tools for depression be presented simply, which would also assist in delivery.

Similarly, findings from the adolescent workshop indicated that for adolescents to be free to request the service, providers need to be accepting, understanding, and non-judgmental. In addition, they asked for depression awareness to be offered to them and their caregivers to improve their mental health knowledge, early detection, and the likelihood of the adolescents receiving support during depression treatment. Finally, caregivers’ awareness was thought necessary to increase help-seeking and treatment adherence and ultimately lead to expected treatment outcomes.

“*For someone like me (adolescent) to benefit from the intervention, my parents need to be aware first to understand my problem and then support me to receive the treatment*.”

Participants in all groups suggested that the CTC intervention should be delivered in HIV-CTCs since its where adolescents receive their regular follow-ups and prescription refill. The Nurse counselor was seen as the most fitting person for the intervention delivery, and most participants believed counselors should be available every clinic day. Adolescents wanted the intervention to be called NITUE. This Swahili word means “Help me offload,” reflecting the intervention’s aim, which is to provide relief from carrying a heavy load of depressive symptoms.

### 9.4. Structure and Delivery of the Session

Adolescents and their caregivers suggested that the experts decide on the number, duration, and frequency of intervention sessions depending on the content and experience delivering similar kinds of help for adolescents living with HIV. However, they preferred that the intervention be provided when needed and not wait until the regular monthly appointments. Health care providers recommended from 30 min to 1 h, as they usually spend during counseling with adherence or behavioral problems. However, they believed that that time would be less if they were appropriately trained and knew what to do; hence, they reached a consensus of 30 min.

“*We have been using so much time because we did not know what to do or say. You might spend an hour sometimes consulting with others or trying to console a crying youth. I think once one is trained and knows what to do, 30 min may be enough*.”

### 9.5. ToC Map

The final Theory of Change map in Figure 1 summarizes a participant’s identification of knowledge gaps and describes pathways to the outcome of interests.

The outcome indicates the primary effect of the intervention. Adolescents and health care providers mentioned improvement in depressive symptoms and enhanced adherence for adolescents living with HIV as the outcome of interest. In addition, all participants indicated other outcomes such as improved academic performance and quality of life as defined by good health, making it easy to enjoy life and attain individual life goals. Stakeholders noted that the final impact of the intervention would be to reduce the treatment gap for adolescent depression.

The intermediate outcome for the intervention included unavailable things and needs to be there to achieve the final prefeed effect. For example, stakeholder and HCP workshops suggested that adequate training, treatment manual, and awareness materials must be available within HIV-CTC. Other intermediate outcomes included (1) improved depression awareness among adolescents, caregivers, and health care providers, (2) decreased mental health stigma among healthcare providers, (3) healthcare providers’ confidence in managing depression improved, (4) adolescents attend CTC regularly for HIV, and depression care, regular screening for depression in routine clinics, (5) enhanced communication skills for providers, and (6) appointments are given for counseling/mental health care.

The interventions discussed were the different components of the intervention that are needed to move from one outcome to another. For example, the unmeant mental health needs were required to establish the gap that the intervention would cover. Healthcare providers suggested that for the nurse counselors to deliver the intervention, there was a need for an intervention to ensure these providers acquire the necessary skills required for screening and managing depression in adolescents. They perceive that refresher, ongoing training, and supervision should enable them to deliver the intervention and give them the skills to provide mental health awareness and care for themselves while helping adolescents. Adolescents suggested that providers need to be understanding and less judgmental for them to be able to ask for and use the intervention. The need for improved communication skills necessitates training as an essential intermediate intervention. Adolescents also perceive that the intervention gives them skills to deal with everyday problems that may lead to depression. Adding problem-solving to the intervention components will make the intervention more desirable and increase help-seeking behavior.

Assumption: Participants also discussed the conditions beyond the interventions that need to be there to move forward in achieving the outcome.

Stakeholders focused on the willingness of nurse counselors to be trained and their desire to deliver the intervention as the fundamental assumption for the intervention’s success. Participants also mentioned sustainable support from the Ministry of Health and other HIV-implementing partners. The HCP workshop discussed issues related to the acceptability of the intervention for adolescents and their willingness to see and receive care. For adolescents, an important assumption was that all adolescents know where and how to ask for mental health services.

Participants were also asked to discuss the rationale from evidence or experience explaining each outcome’s pathway. That is why one outcome is an outcome of another. For example, the available evidence was from a literature review that suggests that task shifting to be effective training is required. Other evidence was from the qualitative studies indicating the initial willingness of providers and positive attitude toward the intervention from adolescents, caregivers, and providers.

Indicators: Participants were asked to list the indicators. These measurable things determine whether the outcome has been achieved or whether there is progress toward achieving the result. For example, participants identified and agreed on the availability of structure, nurse counselors, and youth clubs to indicate that training and awareness education is possible. 

A manual availability will determine whether the training and intervention will occur. Other indicators included (1) change in attitude and improved mental health knowledge from pre and post-test results from training will show if nurse counselors can deliver the intervention, (2) the number of adolescents attending all the intervention sessions, (3) the decrease in depressive symptoms in 3 months as measured by PHQ-A, (4) adolescents’ satisfaction, (5) intervention fidelity by nurse counselors, and (6) changes in depressive symptoms score in intervention vs. Control arm. Figure 1 shows the final Theory of Change map for the NITUE intervention.

## 10. Description of the Intervention Manual

Information from the qualitative study and ToC workshops were merged to identify adolescents’ unmet needs and priorities, which informed the intervention. During the intervention design, the identified needs and preferences were linked to the components of the intervention structured in six sessions.

Session 1: Psychoeducation and engagement to care: The first intervention sessions utilize motivational interviewing and counseling skills to (1) build rapport, (2) provide psychoeducation with a focus on depression, and HIV, as well as available treatment options for depression, (3) provide an explanation of the intervention model of treatment for depression and how it may help with depression and related problems, and (4) set treatment goals (including depression and adherence goals).

### 10.1. Sessions 2–4: Reducing Depressive Symptoms: CBT and Problem-Solving

The three sessions emphasize reducing depressive symptoms using mood monitoring, behavioral activation, cognitive restructuring, and problem-solving techniques. Behavioral techniques such as deep breathing exercises and progressive muscle relaxation are practiced, and adolescents are given homework at each session. Problem solving focuses on the problems adolescents currently face and on facilitating them to discover a way out.

### 10.2. Sessions 5–6: Enhancing Adherence to ART and Relapse Prevention: Taking Charge of One’s Health

Application of brief motivational interviewing strategies to enhance and promote healthy behaviors and help adolescents adhere to ART treatment regimes. It also includes plans on stress management and coping skills.

## 11. Discussion

This study is a comprehensive, systematic, and theory-driven effort to develop the first manualized brief psychological intervention guide in an integrated care context for adolescents living with HIV in Tanzania. The participatory intervention development process was anchored in the theory of change, making it possible to show the causal pathway for how we expect the intervention to achieve the preferred long-term outcome and its impact, as well as the other necessary elements of care preconditions to its success.

We used qualitative exploration first to understand the unmet mental health needs and their associated consequences for adolescents living with HIV and incorporate the perspectives of their caregivers and healthcare providers. The findings of this qualitative data indicated areas that this brief psychological intervention could potentially address. Second, explore challenges to accessing mental health care and psychological intervention implementation opportunities. The understanding of adolescents, their caregivers, and HCP helped to ensure that the developed intervention is acceptable and addresses obstacles to help-seeking and utilization, adding the possibility that the interventions will prove effective. It also made it possible to consider barriers to mental health care access and include strategies to modify those barriers within the structure and content of the intervention [31].

Furthermore, the unmet needs and barriers identified by participants in the qualitative exploration are essential because psychological interventions work through specific factors necessary for the intervention to be effective. Therefore, these components, such as therapeutic alliance, the rationale for treatment, communication skills, positive emotional experience, and locally suitable and adolescent-friendly treatment components, are necessary considerations for intervention development [59].

Thirdly, we used ToC approaches involving different stakeholders, including adolescents living with HIV and their caregivers, HCP, the ministry of health, mental health professionals, and HIV implementing partners. This participatory approach helped the researchers to understand the context of HIV care delivery and make decisions that reflect scientific evidence and the views of essential stakeholders in policymaking, HIV care, and treatment in Tanzania [40,60]. Therefore, the ToC approach merged scientific evidence with stakeholders’ contributions, thus establishing a local buy-in to ensure ownership, acceptance, and support for the intervention from these stakeholders, which are essential prerequisites for the implementation [40]. Furthermore, involving important stakeholders helped build trust, promoting the pooling of resources and knowledge while ensuring the intervention is rooted within the Tanzanian cultural context [60].

Adolescent participants highlighted the importance of involving caregivers in the intervention development process. In addition, literature from LMIC [61] and most mental health treatment guidelines recommend the participation of adolescents and their caregivers in treatment planning and decision-making [17].

Adolescents that live with HIV exhibit high levels of social stigma and isolation from society. Many maintain close contact with a caregiver, parents, or close relatives and perceive family support as critical. Caregivers, therefore, play an essential role in the recovery process. It is also noted that caregivers often establish the adolescent’s initial access to mental health services and that caregiver involvement increases compliance and improves treatment outcomes.

The developed intervention utilizes a task-shifting approach [62] (and uses locally available resources that are accessible within HIV care and treatment facilities [31]. The World Health Organization has recommended integrating mental health services within PHC settings to address the unmet mental health needs of adolescents living with HIV [17,63]. All study participants agreed that nurse counselors in HIV care and treatment facilities to facilitate its future integration within routine HIV care. Although the participants were concerned about the willingness of HCP, nurse counselors who were the targeted providers of the intervention reported willingness and positive attitudes towards the intervention to simplify their work.

All study participants agreed that mental health training and communication for HCP were vital to improving their knowledge, changing negative attitudes about mental illness, and increasing the confidence of nurse counselors to deliver the intervention effectively.

Studies have shown that culturally adapted interventions delivered in a task-shared approach are feasible and acceptable in LMIC [64,65,66]. Findings from poorly resourced countries favor the possibility that HIV counselors in CTC facilities can be successfully trained and supervised to deliver brief psychological interventions that target depression in adolescents living with HIV [67,68,69].

## 12. Strengths and Limitations

We have developed an intervention that may be locally feasible, acceptable, and likely to be effective in reducing depressive symptoms and improving ART adherence for ALWHIV in Tanzania and possibly other low-income countries. In addition to the novel focus, an innovative element includes training resources and a manualized guide of the adapted intervention that targets (1) specific causes of mental health problems and issues facing adolescents living with HIV and (2) management strategies for depression and ART medication adherence problems for adolescents living with HIV and (3) using existing HIV care and treatment providers to deliver the intervention. The study also supports the attainment of Sustainable Development Goal three (Good health and well-being for people), Target 3.3, which requires countries to end AIDS epidemics by 2030, and Target 3.4 to reduce by one-third premature mortality from non-communicable diseases (suicide and depression included) through prevention, treatment, and promotion of mental health and well-being by the same year 2030.

## 13. Conclusions

Utilizing the TOC-MRC enhanced framework for complex intervention development, we developed a culturally appropriate brief psychological intervention that uses a task-shifting approach to address depression and medication adherence for ALWHIV in Dar es Salaam, Tanzania. The intervention will be piloted for appropriateness, feasibility, and acceptability. The study also provides the prerequisite materials, training, and infrastructure needed for a future trial to determine the effectiveness of this intervention in reducing depressive symptoms and enhancing ART adherence in adolescents living with HIV in Tanzania and possibly other countries in sub-Saharan Africa.

## Figures and Tables

**Figure 1 healthcare-10-02491-f001:**
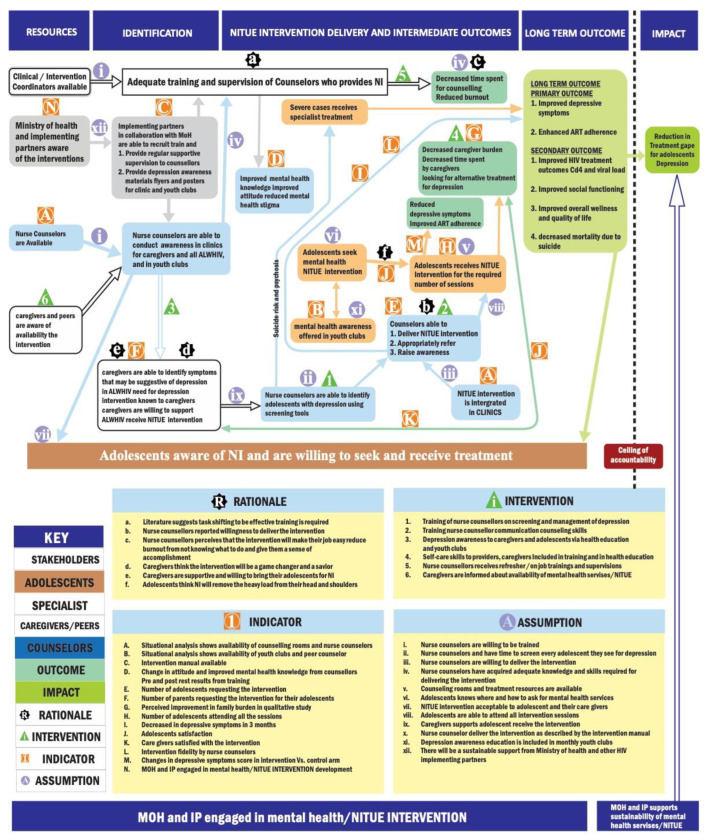
Theory of Change Map for the NITUE Intervention.

**Table 1 healthcare-10-02491-t001:** Description of study participants.

Participants	Male	Female	Total
ToC with Stake Holders			
HIV implementing partners	-	4	4
MoH and government officials	3	-	3
ToC with adolescent	4	4	8
ToC with healthcare providers	3	4	7
Total			22
Representative meeting			
Stake Holders	1	2	7
Adolescent	2	2	

## Data Availability

All data generated in this study are available from the corresponding author upon reasonable request.

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
