# Peer review of "Development of a Psychological Intervention to Improve Depressive Symptoms and Enhance Adherence to Antiretroviral Therapy among Adolescents and Young People Living with HIV in Dar es Salaam Tanzania"

_healthcare, 2022, doi:10.3390/healthcare10122491_

Round 1

Reviewer 1 Report

The work is unprecedented and relevant. The method must include ethical issues involving the steps with participants. This is a methodological error and must be corrected. The authors designed an intervention for adolescents living with HIV based on evidence and the participation of the various actors involved in this assistance, however, the intervention was not tested. Therefore, its effectiveness cannot be proved. The editor must be aware of this search limitation.

Author Response

PLEASE SEE THE ATTACHMENT,

THANK YOU.

Reviewer 2 Report

1) The authors used Theory of Change enhanced Medical Research Council. It is better if the authors can describe the theory in a bit details and how the theory of change fit to this study.

2) The authors are advised to add more recent literature regarding the effective interventions to improve depression. Applying Western approaches and interventions that fit to other cultures are also not recent. 

3) Please check the referencing style because it is not consistent. Some to the references do not provide year published. 

Author Response

Please see the attachment,

Thank you.
